# Grain Yield and Quality of Winter Wheat Depending on Previous Crop and Tillage System

**Dorota Gawęda**  **and Małgorzata Haliniarz \*** 

Department of Herbology and Plant Cultivation Techniques, University of Life Sciences in Lublin, Akademicka 13, 20-950 Lublin, Poland; dorota.gaweda@up.lublin.pl
\* Correspondence: malgorzata.haliniarz@up.lublin.pl; Tel.: +48-81-4456654

**Abstract:** The effects of previous crops (soybean (*Glycine max* (L.) Merr.) and winter oilseed rape (*Brassica napus* L. ssp. *oleifera* Metzg)), as well as of conventional tillage (CT) and no-tillage (NT), on yield and some quality parameters of winter wheat (*Triticum aestivum* L.) grain were evaluated based on a four-year field experiment. Wheat was grown in a four-field crop rotation: Soybean—winter wheat—winter oilseed rape—winter wheat. The study revealed that growing winter wheat after soybean, compared to its cultivation in the field after winter oilseed rape, significantly increased grain and straw yield, as well as all yield and crop components evaluated. After the previous soybean crop, higher grain protein content, Zeleny sedimentation value, and grain uniformity were also found. After winter oilseed rape, only a greater value of the gluten index was obtained. Statistical analysis did not show the tillage system (TS) to influence the grain yield of winter wheat. Under the CT system, relative to NT, straw yield, number of ears per 1 $m^2$, and plant height of winter wheat were found to be significantly higher. The NT system, on the other hand, beneficially affected the thousand grain weight. Wheat grain harvested under the CT system was characterized by a higher grain test weight, better grain uniformity, and lower gluten index than under NT.

**Keywords:** previous crop; tillage system; grain quality; yield; *Triticum aestivum* L.



## 1. Introduction

Wheat (*Triticum aestivum* L.), including its winter form, is the most important crop in the world alongside rice (*Oryza sativa* L.), maize (*Zea mays* L.), and soybean (*Glycine max* (L.) Merr.) [1]. Wheat grain is intended for human consumption and for animal feed. After its processing, it is used to produce flour, groats, cereals, pasta, and bakery products or as an additive to other food and animal feed products [2–4].

The economic value of individual winter wheat cultivars is determined by their yield quality and quantity. These parameters allow the possible use of grain to be determined. Grain intended for consumption purposes should be characterized by high baking and milling value (processing value). Therefore, it is not only the high yield that counts in wheat production but also favorable grain quality parameters, among others, as follows: Protein content, gluten quantity and quality, sedimentation value, and grain test weight. In Poland, winter wheat is predominantly used to make flour and to bake bakery products. The value of flour is greatly affected by grain quality, which is largely genetically determined [5]. Nonetheless, selecting a cultivar with favorable characteristics does not automatically guarantee that the yield of desired quality will be obtained, as the yielding potential of crop plants can be fully utilized after they have been provided with optimal growth conditions [6,7]. This goal can be achieved by, among others, selecting an appropriate position in the crop rotation and proper agronomic practices. This is of essential importance for the quantity and reliability of obtained yields, which consequently affects indirectly the quality of grain and its commercial value [8–10].

Winter wheat is a crop with high habitat and agronomic requirements, particularly sensitive to an unfavorable previous crop. This species should be sown after, among

others, oilseed rape (*Brassica napus* L. ssp. *oleifera* Metzg) that shades well the soil and leaves a lot of crop residue. The potatoes (*Solanum tuberosum* L.) and legumes are also appropriate previous crops for wheat [11,12]. Studies of many authors reveal beneficial production effects after legumes used as a previous crop [12–16]. However, wheat is most frequently sown after cereals, which results in decreased yield and worse grain quality due to increased weed infestation [17] and greater plant infection by pathogens, causing stem base diseases [18,19]. A decrease in volumetric weight and homogeneity of grain, as well as reduced wet gluten content in winter wheat grain, among others, can be observed in such a stand [20].

Tillage also has a significant effect on grain quantity and quality as, by modifying the soil physical, chemical, and biological properties, it directly impacts the growth and development of crops [10,21–25]. When selecting the tillage system, we try to create optimal conditions for producing a high yield of grain with favorable quality parameters. Nevertheless, the opinions on the effect of tillage systems on crop yield are ambiguous because the obtained results largely depend on habitat conditions and the crop species [26–33]. When selecting the tillage system, the economic aspect should also be considered by taking into account the impact of tillage on soil properties. The fact is that wheat cultivation in the conventional system can increase production costs and make the soil compact, hence causing its insufficient aeration [34,35]. Destruction of the topsoil structure and decreased biological diversity of soil are also the reasons for abandonment of conventional tillage [36,37].

The growing human population and the universal use of cereal grain for food and animal feed purposes create the need to increase its production [38,39]. Among cereals, winter wheat grain is a strategic raw material in the global food economy, and that is why research aimed at determining optimal agronomic conditions is of key importance for obtaining this crop's high yields with favorable quality parameters. Taking into account the fact that the tillage system and selection of an appropriate crop rotation position can significantly influence crop yield quantity and quality, the present study was conducted to investigate these issues. Its aim was to evaluate the effects of previous crops (soybean and winter oilseed rape), as well as of conventional tillage (CT) and no-tillage (NT), on the yield and some quality parameters of winter wheat grain.

## 2. Materials and Methods

### 2.1. Location of the Experiment and Soil and Climatic Conditions

This experiment was carried out over the period 2014–2017 at the Uhrusk Experimental Farm in Poland, belonging to the University of Life Sciences in Lublin (51°18′ N, 23°36′ E), on Rendzic Phaeozem [40] with a grain-size distribution of sandy loam. The soil arable layer was characterized by alkaline pH (in 1 M KCl = 7.7), very high phosphorus availability (229.8 mg P kg$^{-1}$ soil), high potassium availability (150.2 mg K kg$^{-1}$ soil), and very low magnesium availability (16 mg Mg kg$^{-1}$ soil). The humus content was 1.5%, while the content of fine particles (<0.02 mm) in the 0–30 cm layer was 20.7%.

Throughout the duration of the experiment, lower total precipitation than the long-term average was recorded in the years 2014/2015 and 2016/2017 (Table 1). During the 2014/2015 growing season, the total precipitation was lower by 187 mm than the long-term average for the same period. However, the least precipitation occurred during the wintering period of wheat (February) and in the grain harvest month (August), and therefore, it did not have a major impact on yield of this cereal crop. The highest amount of precipitation occurred during the third growing season of winter wheat (2015/2016). The highest precipitation was recorded in the wheat sowing month (September) and during grain ripening (July). In the first growing season of winter wheat (2013/2014), the total precipitation was also found to be higher than the long-term average, particularly in the month of May.

In all experimental years, a higher temperature was recorded during the growing season of winter wheat (September–August) than the long-term average for these months (Table 1). Throughout the duration of the experiment, the temperature values were favor-

able in the months from March to August during the period of intensive wheat growth and grain ripening.

**Table 1.** Total precipitation and mean monthly air temperature in the growing season of winter wheat, recorded by the Meteorological Station in Bezek (Poland).

| Months | Years | | | | | | | | | |
|---|---|---|---|---|---|---|---|---|---|---|
| | **2013/2014** | | **2014/2015** | | **2015/2016** | | **2016/2017** | | **LTA * 1974–2010** | |
| | **Precipitation (mm)/Temperature (°C)** | | | | | | | | | |
| | **mm** | **°C** | **mm** | **°C** | **mm** | **°C** | **mm** | **°C** | **mm** | **°C** |
| September | 89.8 | 11.6 | 45.4 | 13.9 | 94.2 | 16.6 | 11.0 | 15.0 | 57.1 | 12.9 |
| October | 3.1 | 10.0 | 30.3 | 9.3 | 44.3 | 6.9 | 120.7 | 6.8 | 40.8 | 7.7 |
| November | 71.9 | 5.6 | 15.1 | 4.2 | 46.7 | 6.6 | 17.0 | 2.3 | 32.3 | 2.4 |
| December | 7.4 | 1.0 | 40.9 | −0.2 | 25.8 | 3.5 | 15.3 | −0.1 | 29.7 | −1.7 |
| January | 55.5 | −2.7 | 32.0 | 0.6 | 38.7 | −4.2 | 6.0 | −5.2 | 23.6 | −3.2 |
| February | 8.1 | 1.3 | 2.6 | 0.3 | 52.4 | 3.2 | 42.8 | −1.7 | 21.8 | −2.5 |
| March | 30.2 | 6.1 | 38.5 | 4.7 | 52.1 | 3.8 | 30.9 | 5.8 | 26.8 | 1.7 |
| April | 44.0 | 9.3 | 33.7 | 7.7 | 67.7 | 9.3 | 59.5 | 7.4 | 37.9 | 7.8 |
| May | 151.6 | 13.9 | 62.2 | 13.1 | 54.5 | 14.9 | 71.6 | 14.2 | 57.4 | 13.5 |
| June | 88.2 | 15.8 | 15.5 | 17.1 | 66.4 | 18.1 | 27.0 | 17.9 | 76.9 | 16.3 |
| July | 35.9 | 20.8 | 45.4 | 21.7 | 131.5 | 20.0 | 99.5 | 20.1 | 81.6 | 18.2 |
| August | 85.2 | 18.8 | 6.9 | 22.2 | 53.4 | 18.9 | 39.3 | 20.1 | 69.8 | 17.6 |
| Sum/Mean (September–August) | 670.9 | 9.3 | 368.5 | 9.6 | 727.7 | 9.8 | 540.6 | 8.6 | 555.7 | 7.6 |

* LTA—long term average.

The data presented in Table 1 show that similar weather conditions were observed in all experimental years. The thermal conditions were favorable, whereas the amount of precipitation was sufficient for the growth and yield of winter wheat.

*2.2. Experimental Design and Agronomic Practices*

The experiment on winter wheat was set-up in a randomized block design with three replicates in 32 m$^2$ plots. The study object was the winter wheat cultivar "Astoria", which is recorded in Poland as an elite, highest quality wheat cultivar (group E) with a very good genetically determined processing quality of grain for bread baking. The experimental factors were as follows:

I.    PC—previous crop of winter wheat: Soybean, winter oilseed rape;
II.   TS—tillage system: CT—conventional tillage; NT—no-tillage.

In the conventional system (CT), tillage for wheat sown after soybean involved the following: Pre-sowing ploughing, harrowing, grain sowing, and post-sowing harrowing. In the spring, harrowing was done twice. Under no-tillage (NT), after the harvest of soybean, the following soil management operations were carried out: Tilling with a stubble cultivator (grubber + cage roller) instead of ploughing, grain sowing, and post-sowing harrowing, while in the spring, the same agronomic operations were carried out as under the conventional tillage system. After harvest of the winter oilseed rape crop, skimming was additionally done in the conventional tillage treatment, whereas under no-tillage conditions, a stubble cultivator (grubber + cage roller) was used instead of skimming. Further treatments were the same as in the field after soybean. Wheat was grown in a four-field crop rotation: Soybean—winter wheat—winter oilseed rape—winter wheat, which was conducted in all the fields simultaneously. In autumn 2012, winter wheat and winter oilseed rape were sown, and soybeans were sown in spring 2013. These plants were cultivated in a conventional tillage and no-tillage system, which was continued in the following years. The results presented in the work concern the growing seasons 2013/2014–2016/2017, in which winter wheat was sown after the previous crops planned

in the experiment scheme. This allowed for the evaluation of the four-year use of previous crops and tillage system for winter wheat.

Every year, mineral fertilizers were applied at the following rates: N—120 kg ha$^{-1}$ as ammonium nitrate NH$_4$NO$_3$ in dose 353 kg ha$^{-1}$ (Pulan$^{®}$, Grupa Azoty Puławy, N—34%), P—60 kg ha$^{-1}$ as superphosphate in dose 345 kg ha$^{-1}$ (Super Fos Dar 40, Gdańskie Zakłady Nawozów Fosforowych "Fosfory" Sp. z o.o., P$_2$O$_5$—40%), K—90 kg ha$^{-1}$ as potassium chloride in dose 181 kg ha$^{-1}$ (Potassium salts, Luvena S.A., K$_2$O—60%). Phosphorus and potassium fertilizers were applied in whole before sowing. Nitrogen fertilizers were used three times during the growing season of wheat. The first part of the dose at an amount of 50 kg ha$^{-1}$ was applied in the spring, right at the beginning of the growing season (BBCH 22–23) [41]. The other two doses of nitrogen fertilizers, at an amount of 35 kg h$^{-1}$ each dose, were applied during stem elongation (BBCH 32) and at the beginning of heading (BBCH 51), respectively.

Every year (during the period 2013–2016), winter wheat was sown in the last 10 days of September. Grain was sown at a rate of 5.5 million grains ha$^{-1}$. Before sowing, the seed dressing Sarfun T 65 DS (a.i. thiuram 45%, carbendazim 20%, Zakłady Chemiczne 'Organika-Sarzyna' S.A.) was applied at a rate of 200 g per 100 kg of grain with 800 mL water addition.

In the spring at the very beginning of the growing season (BBCH 23), the herbicide Lancet Plus 125 WG (a.i. aminopyralid 5.0%, pyroxsulam 5.0%, florasulam 2.5%, Dow AgroSciences Polska Sp. z o.o) was sprayed at a rate of 0.2 kg ha$^{-1}$. At the first node stage (BBCH 31), the fungicide Alert 375 SE (carbendazim 250 g L$^{-1}$, flusilazole 125 g L$^{-1}$, Du Pont Poland Sp. z o.o.) was applied at a rate of 1 L ha$^{-1}$, while at the beginning of heading (BBCH 51), Tilt Turbo 575 EC (fenpropidin 45.9%, propiconazole 12.8%, Syngenta Polska Sp. z o.o.) was applied at a rate of 1 L ha$^{-1}$. When a threat from aphids occurred, Decis 2.5 EC (deltamethrin 2.8%, Bayer Sp. z o.o.) was applied at full heading (BBCH 55) at a rate of 0.25 L ha$^{-1}$.

Every year (during the period 2014–2017), winter wheat was harvested in the first 10 days of August.

### 2.3. Methods of Plant Analyses

Grain yield (Mg ha$^{-1}$), straw yield (Mg ha$^{-1}$), and the following yield and crop components of winter wheat: Ear density (ears m$^{-2}$), plant height (cm), number of grains per ear (grains), grain weight per ear (g), and thousand grain weight (g), were the study object. The following quality characteristics of winter wheat grain were also determined: Total protein content (%), wet gluten content (%), gluten index (%), Zeleny sedimentation value (mL), grain test weight (kg hL$^{-1}$), and grain uniformity (%).

Grain yield and straw yield of winter wheat were weighed separately for each plot, and the obtained results were expressed on a per hectare basis. Number of grains per ear and grain weight per ear were determined based on a sample consisting of 30 ears randomly selected from each plot. Ear density per 1 m$^2$ was determined before the harvest of winter wheat in two randomly selected places in each plot, delineated by a 1 m × 0.25 m quadrat frame. Thousand grain weight was calculated after wheat harvest (2 × 500 grains from each plot).

Total nitrogen was determined by the Kjeldahl method and expressed as a percentage protein content. Grain wet gluten content and gluten index (PN-EN ISO 21415-2:2008) were calculated using the Glutomatic 2200 system. Grain test weight was determined on grain test weight scales using the standard hectoliter weight apparatus according to the standard PN-73/R-740. Grain uniformity (the weight of grains remaining on a 2.5 × 25 mm mesh screen) was evaluated according to the standard BN-69/9131-02. Zeleny sedimentation value was determined according to the standard PN—ISO 5529.

*2.4. Statistical Analysis*

The study results collected over the period 2014–2017 were analyzed by analysis of variance (ANOVA), while the significance of differences was estimated by Tukey's test at a significance level of <0.05. ANALWAR-5.3.FR statistical software was used for calculations. The results were subjected to 2-way analysis of variance in a randomized block design and synthesis of the data was made according to the mixed model. The effect of previous crops and tillage systems and their interaction on grain yield, straw yield, and quality parameters of winter wheat grain were determined. Significant differences between the research years and the interaction of the years and experimental factors were not found. For this reason, these data are not shown in this manuscript. Furthermore, to determine dependencies and relationships between the selected characteristics (yield and its components), Pearson's linear correlation coefficient was established, a linear regression analysis was performed, and the coefficient of determination ($R^2$) was determined. Statistica 13.1 software was employed for the above-mentioned calculations.

## 3. Results

*3.1. Yield and Yield Components of Winter Wheat*

After the previous soybean crop, the winter wheat grain yield was 5.84 Mg ha$^{-1}$, being 9.6% higher than that after winter oilseed rape (Figure 1). In the field after soybean, all the investigated yield and crop components of winter wheat were also found to be more favorable. After the previous winter oilseed rape crop, the ear density per 1 m$^2$, plant height, grain number and weight per ear, and thousand grain weight were lower by 2.6, 2.9, 6.3, 8.4, and 2.3%, respectively, compared to those found in the field after soybean (Table 2).

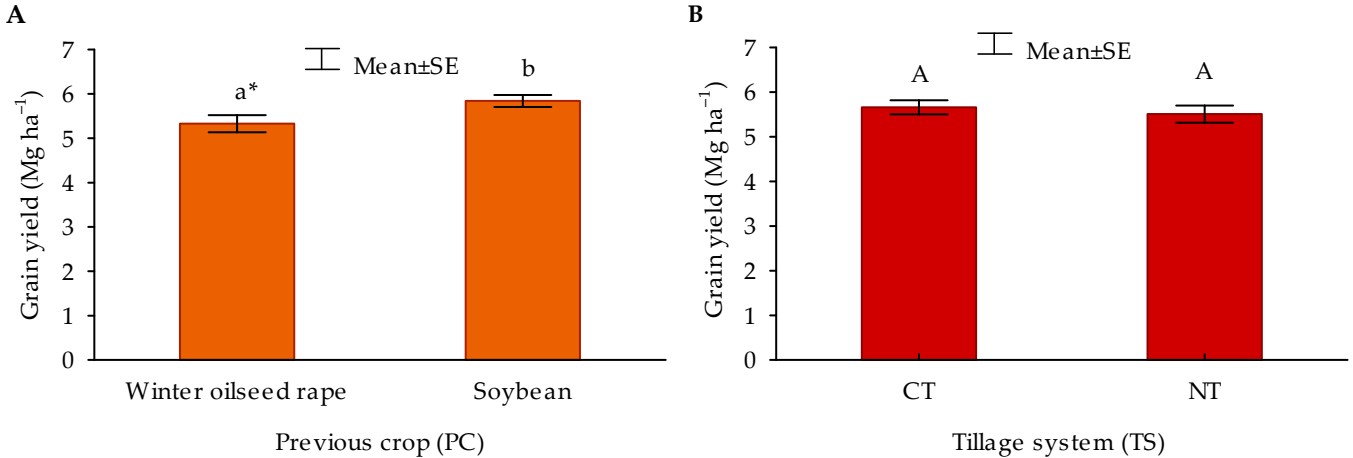

**Figure 1.** Winter wheat grain yield depending on previous crop (**A**) and tillage system (**B**) (mean for 2014–2017). CT—Conventional tillage; NT—No-tillage. * Different letters denote significant differences ($p \leq 0.05$). The same letter means not significantly different values. The lowercase letters refer to previous crops (PC); the capital letters refer to tillage systems (TS).

Tillage system (TS) did not cause significant differences in grain yield of winter wheat, but its increasing trend was found for the CT system relative to NT (Figure 1). In the conventional system (CT), the number of ears per 1 m$^2$ was proven to be higher by 3.6%, while the plant height was higher by 2.5 cm. Under the NT system, on the other hand, the thousand grain weight was found to be greater by 1.7% than for CT. In the case of both tillage systems (CT and NT), the number and weight of grains per ear of winter wheat were found to be similar (Table 2).

The statistical analysis did not show an interaction between previous crop and tillage system in determining winter wheat grain yield (Figure 2). In the treatment with plough tillage, the grain yield was only slightly higher after the previous soybean crop than in the other experimental treatments.

**Table 2.** Winter wheat yield and crop components depending on previous crop and tillage system (mean for 2014–2017).

| Specification | Previous Crop (PC) | | *p*-Value | Tillage System (TS) | | *p*-Value |
|---|---|---|---|---|---|---|
| | Winter Oilseed Rape | Soybean | | CT | NT | |
| Number of ears (no. m$^{-2}$) | 459.5 a [1] | 471.7 b | 0.0383 * | 473.8 B | 457.3 A | 0.0065 ** |
| Plant height (cm) | 80.3 a | 82.7 b | 0.0318 * | 82.7 B | 80.2 A | 0.0250 * |
| Number of grains per ear (no.) | 25.3 a | 27.0 b | 0.0000 *** | 26.1 A | 26.2 A | 0.8212 |
| Grain weight per ear (g) | 1.20 a | 1.31 b | 0.0008 *** | 1.23 A | 1.27 A | 0.2158 |
| Thousand grain weight (g) | 45.8 a | 46.9 b | 0.0041 ** | 45.9 A | 46.7 B | 0.0397 * |

CT—Conventional tillage; NT—No-tillage. [1] Different letters denote significant differences ($p \leq 0.05$). The same letter means not significantly different values (values for horizontal lines). The lowercase letters refer to previous crops (PC); the capital letters refer to tillage systems (TS). * significance level at $p \leq 0.05$, ** significance level at $p \leq 0.01$, *** significance level at $p \leq 0.001$.

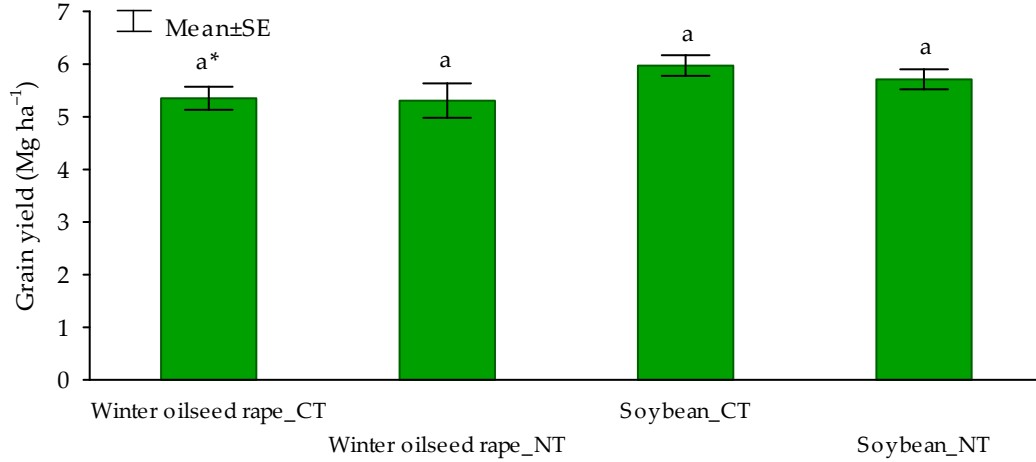

**Figure 2.** Interactive dependencies of previous crop and tillage system in determining winter wheat grain yield (mean for 2014–2017). CT—Conventional tillage; NT—No-tillage. * The same letter means not significantly different values ($p \leq 0.05$).

The interaction of previous crop (PC) and tillage system (TS) was proven to affect the number of grains per ear and thousand grain weight of winter wheat (Table 3). The highest number of grains per ear was obtained in the field after soybean under CT, while it was significantly lower after the previous winter oilseed rape crop both under CT (by 8.8%) and NT (by 5.8%). The highest thousand grain weight was found when wheat was sown after soybean under NT conditions. Compared to this treatment, the thousand grain weight was much lower after oilseed rape under the NT and CT systems, as well as after soybean under CT, respectively, by 4.0%, 3.8%, and 3.2%.

**Table 3.** Interactive dependencies of previous crop and tillage system in determining winter wheat yield and crop components (mean for 2014–2017).

| Specification | Wheat after Oilseed Rape | | Wheat after Soybean | | *p*-Value |
|---|---|---|---|---|---|
| | CT | NT | CT | NT | |
| Number of ears (no. m$^{-2}$) | 465.1 a [1] | 453.8 a | 482.5 a | 460.9 a | 0.3711 |
| Plant height (cm) | 80.9 a | 79.6 a | 84.6 a | 80.7 a | 0.2334 |
| Number of grains per ear (no.) | 24.9 a | 25.7 ab | 27.3 c | 26.7 bc | 0.0369 * |
| Grain weight per ear (g) | 1.17 a | 1.23 a | 1.30 a | 1.32 a | 0.5807 |
| Thousand grain weight (g) | 45.8 a | 45.7 a | 46.1 a | 47.6 b | 0.0244 * |

CT—Conventional tillage; NT—No-tillage. [1] Different letters denote significant differences ($p \leq 0.05$). The same letter means not significantly different values. * significance level at $p \leq 0.05$.

Both experimental factors substantially modified winter wheat straw weight (Figure 3). The straw yield was higher by 11.0% after the previous soybean crop than after winter oilseed rape and by 7.5% under CT in comparison with NT.

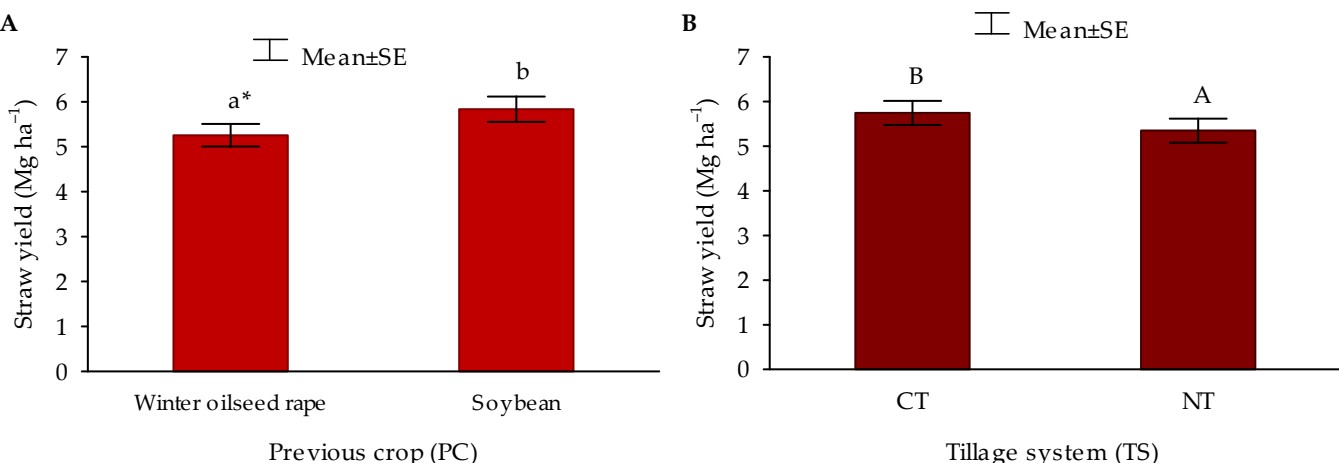

**Figure 3.** Winter wheat straw yield depending on previous crop (**A**) and tillage system (**B**) (mean for 2014–2017). CT—Conventional tillage; NT—No-tillage. * Different letters denote significant differences ($p \leq 0.05$). The lowercase letters refer to previous crops (PC); the capital letters refer to tillage systems (TS).

The interaction between previous crop and tillage was not shown to impact the straw yield of winter wheat (Figure 4). Under CT conditions, the straw weight was only slightly higher after the previous soybean crop compared to the other experimental treatments.

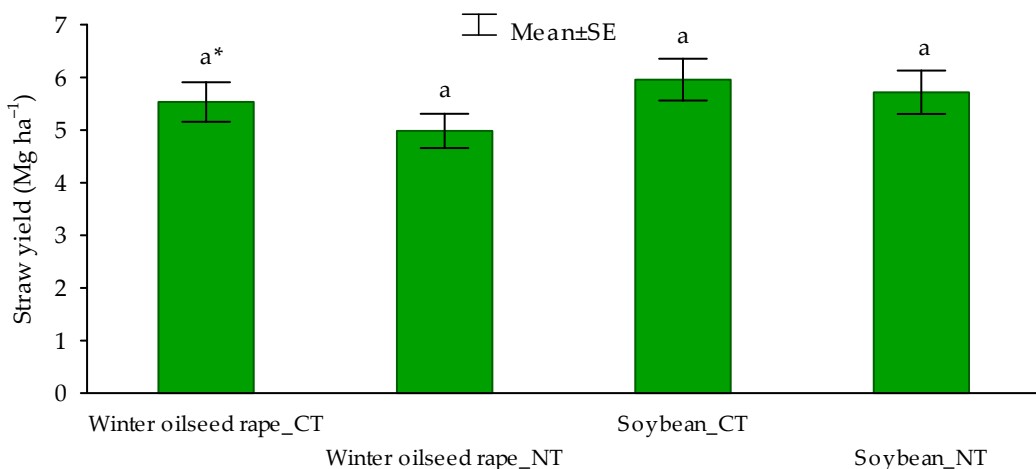

**Figure 4.** Interactive dependencies of previous crop and tillage system in determining winter wheat straw yield (mean for 2014–2017). CT—Conventional tillage; NT—No-tillage. * The same letter means not significantly different values ($p \leq 0.05$).

The grain yield of winter wheat grown after winter oilseed rape was significantly positively correlated with ear density before harvest, number and weight of grains per ear, and 1000 grain weight (Table 4). The regression correlation model demonstrates that an increase in this trait by one unit caused an increase in yield by about 0.01 Mg ha$^{-1}$. The value of the coefficient of determination ($R^2$) shows that about 43% of the variation in yield is explained by the presented model, whereas the correlation coefficient of 0.66 indicates a moderately strong relationship between the variables. A 1 g increase in grain weight per ear resulted in the highest average increase in yield, which was 2.40 Mg ha$^{-1}$. The $R^2$ indicates that about 33% of the variation in yield is explained by the presented model, whereas the correlation coefficient of 0.57 reveals a strong relationship between the variables. An increase in the other traits by one unit contributed to an increase in yield ranging from 0.08 to 0.28 Mg ha$^{-1}$. Yield of winter wheat grown after soybean was significantly positively influenced by ear density before harvest, straw weight, plant height, and number of grains per ear. An increase in these traits by one unit resulted in a rise

in yield from about 0.01 to 0.27 Mg ha$^{-1}$. The correlation coefficients reveal a moderate relationship between yield and the analyzed characteristics.

**Table 4.** Simple correlation coefficients, probability of significance, and simple regressions for the relationship between winter wheat yield and yield components depending on previous crop.

| Quality Parameters | $r_{emp}$ | $p$ | $R^2$ | Regression Equation |
|---|---|---|---|---|
| | Wheat after oilseed rape | | | |
| Ear density before harvest | 0.66 | 0.000 *** | 0.43 | y = 0.65014 + 0.0101829x |
| Number of grains per ear | 0.52 | 0.010 ** | 0.27 | y = 2.08681 + 0.12806x |
| Grain weight per ear | 0.57 | 0.003 ** | 0.33 | y = 2.44599 + 2.40362x |
| TGW | 0.49 | 0.014 * | 0.25 | y = −3.31825 + 0.188979x |
| | Wheat after soybean | | | |
| Ear density before harvest | 0.65 | 0.000 *** | 0.43 | y = 2.41872 + 0.00725778x |
| Straw weight | 0.57 | 0.004 ** | 0.32 | y = 4.24591 + 0.273364x |
| Plant height | 0.54 | 0.007 ** | 0.29 | y = 3.4077 + 0.0294402x |
| Number of grains per ear | 0.41 | 0.046 * | 0.17 | y = 3.54405 + 0.0850837x |

$r_{emp}$—Pearson's correlation coefficient, $R^2$—coefficient of determination, TGW—thousand grain weight, * significance level at $p \leq 0.05$, ** significance level at $p \leq 0.01$, *** significance level at $p \leq 0.001$.

The dependence of winter wheat grain yield on yield components was different for the tillage systems studied (Table 5). In the case of the conventional tillage system, it was shown that with an increase in ear density before harvest, straw weight, and plant height by one unit, the yield increased on average from 0.01 to 0.36 Mg ha$^{-1}$. Under no-tillage, ear density before harvest, grain number and weight per ear, and 1000 grain weight had a significant effect on winter wheat yield. The linear regression correlation model shows that a 1 g increase in this trait resulted in an average increase in grain yield by as much as about 2.91 Mg ha$^{-1}$. The value of the coefficient of determination ($R^2$) indicates that about 50% of the variation in yield is explained by the presented model. The correlation coefficient of 0.71 demonstrates a moderately strong relationship between these traits.

**Table 5.** Simple correlation coefficients, probability of significance, and simple regressions for the relationship between winter wheat yield and yield components under the different tillage systems.

| Quality Parameters | $r_{emp}$ | $p$ | $R^2$ | Regression Equation |
|---|---|---|---|---|
| | CT—Conventional tillage | | | |
| Ear density before harvest | 0.69 | 0.000 *** | 0.48 | y = 1.62073 + 0.00852792x |
| Straw weight | 0.63 | 0.002 ** | 0.39 | y = 3.57226 + 0.363493x |
| Plant height | 0.60 | 0.002 ** | 0.36 | y = 2.71961 + 0.0355507x |
| | NT—No-tillage | | | |
| Ear density before harvest | 0.61 | 0.002 ** | 0.37 | y = 1.05388 + 0.00974278x |
| Number of grains per ear | 0.61 | 0.002 ** | 0.38 | y = 1.68786 + 0.145875x |
| Grain weight per ear | 0.71 | 0.000 *** | 0.50 | y = 1.81808 + 2.90732x |
| TGW | 0.41 | 0.047 * | 0.17 | y = −0.207055 + 0.122523x |

$r_{emp}$—Pearson's correlation coefficient, $R^2$—coefficient of determination, TGW—thousand grain weight, * significance level at $p \leq 0.05$, ** significance level at $p \leq 0.01$, *** significance level at $p \leq 0.001$.

### 3.2. Quality Parameters of Winter Wheat Grain

Total protein content in winter wheat grain differed significantly depending on the previous crop (PC) (Figure 5). A much higher amount of this component was found after the previous soybean crop (13.0%) than after winter oilseed rape (12.6%). Tillage system (TS) was not proven to influence protein content in wheat grain.

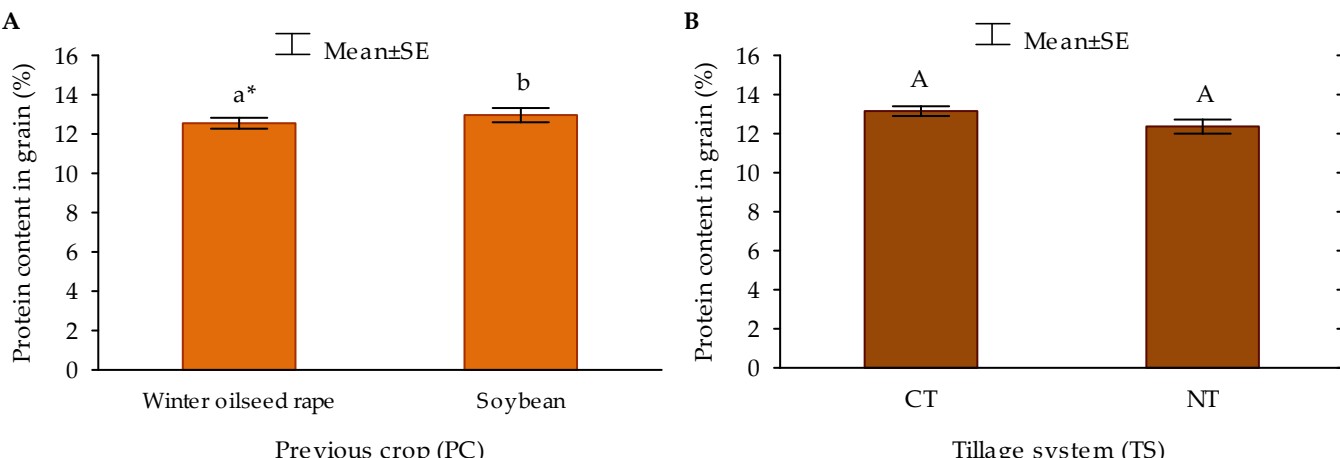

**Figure 5.** Total protein content in winter wheat grain depending on previous crop (**A**) and tillage system (**B**) (mean for 2014–2017). CT—Conventional tillage; NT—No-tillage. * Different letters denote significant differences ($p \leq 0.05$). The same letter means not significantly different values. The lowercase letters refer to previous crops (PC); the capital letters refer to tillage systems (TS).

Tillage system did not cause significant differences in wet gluten content in winter wheat grain (Figure 6). The percentage content of this component was also at a similar level (28.1% and 28.5%) after both previous crops.

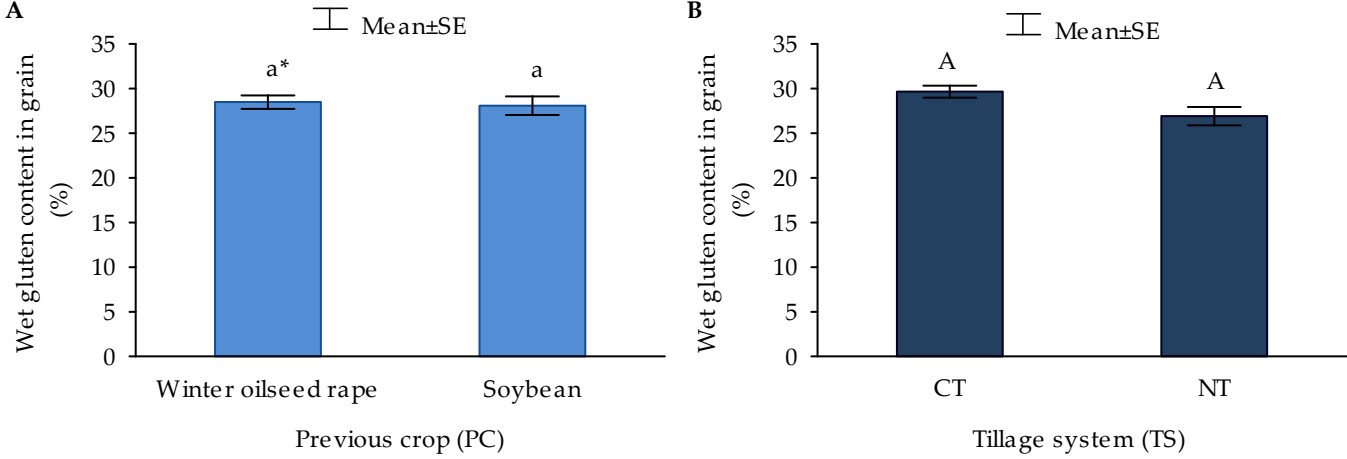

**Figure 6.** Wet gluten content in winter wheat grain depending on previous crop (**A**) and tillage system (**B**) (mean for 2014–2017). CT—Conventional tillage; NT—No-tillage. * The same letter means not significantly different values ($p \leq 0.05$). The lowercase letters refer to previous crops (PC); the capital letters refer to tillage systems (TS).

The analysis of variance revealed a significant effect of previous crop and tillage system (TS) on the gluten index of winter wheat grain (Figure 7). A substantially higher value of the parameter in question was found after the previous winter oilseed rape crop (81.5%) than in the field after soybean (80.2%), as well as under NT conditions (83.5%) relative to CT (78.2%). In all research objects, gluten can be assessed as strong.

In the field after soybean, a significantly higher Zeleny sedimentation value was shown than after the previous winter oilseed rape crop, with this difference being 2.0 mL (Figure 8). Under the CT and NT systems, the sedimentation index reached a similar value.

Grain test weight, characterizing grain size and filling, was largely dependent on tillage system (Figure 9). Grain harvested from the conventional tillage treatment was characterized by a significantly higher value of the characteristic in question compared to that obtained for the NT system. After both previous crops, the grain test weight had a similar value (76.8 and 77.2 kg hL$^{-1}$, respectively).

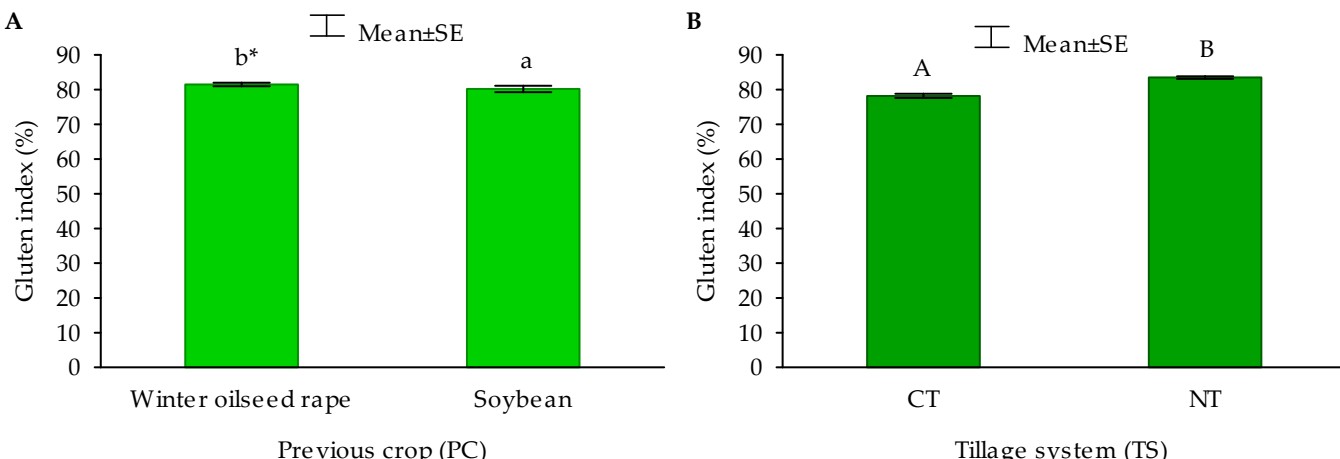

**Figure 7.** Gluten index for winter wheat grain depending on previous crop (**A**) and tillage system (**B**) (mean for 2014–2017). CT—Conventional tillage; NT—No-tillage. * Different letters denote significant differences ($p \leq 0.05$). The lowercase letters refer to previous crops (PC); the capital letters refer to tillage systems (TS).

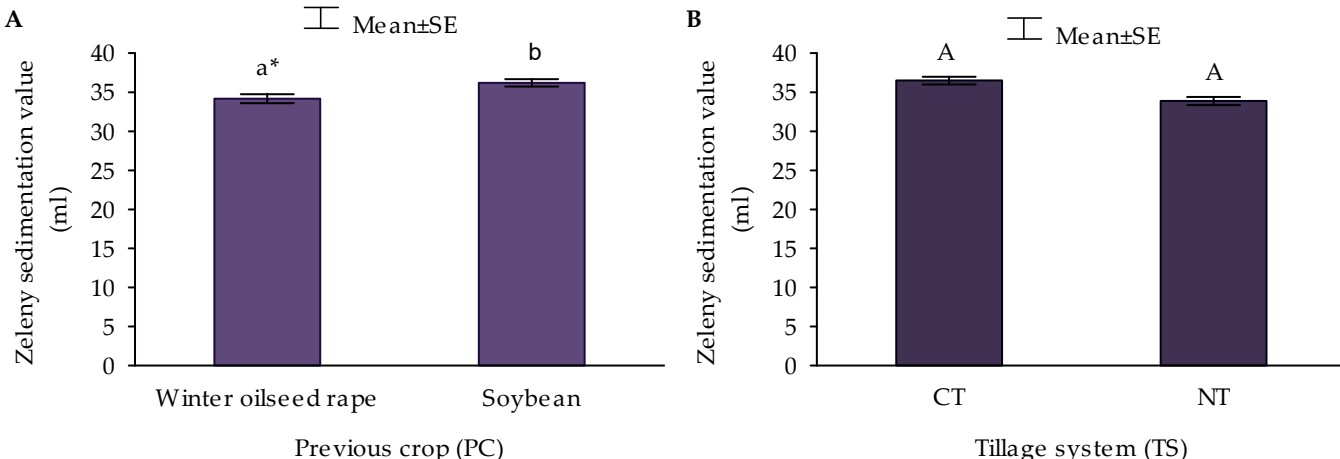

**Figure 8.** Zeleny sedimentation value for winter wheat grain depending on previous crop (**A**) and tillage system (**B**) (mean for 2014–2017). CT—Conventional tillage; NT—No-tillage. * Different letters denote significant differences ($p \leq 0.05$). The same letter means not significantly different values. The lowercase letters refer to previous crops (PC); the capital letters refer to tillage systems (TS).

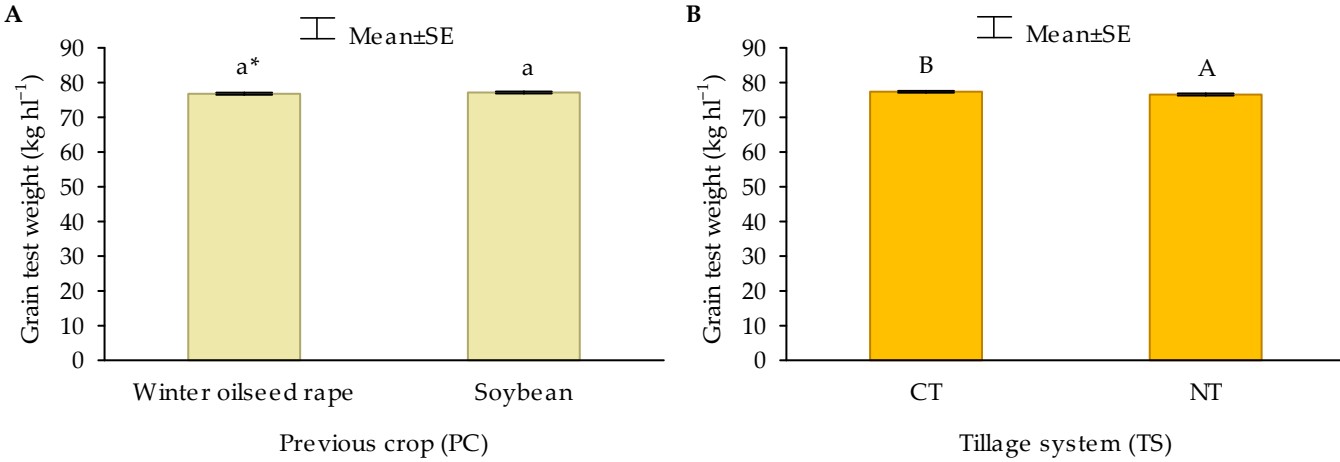

**Figure 9.** Test weight of winter wheat grain depending on previous crop (**A**) and tillage system (**B**) (mean for 2014–2017). CT—Conventional tillage; NT—No-tillage. * Different letters denote significant differences ($p \leq 0.05$). The same letter means not significantly different values. The lowercase letters refer to previous crops (PC); the capital letters refer to tillage systems (TS).



Both experimental factors (PC and TS) significantly modified the grain uniformity of winter wheat (Figure 10). The characteristic in question was found to have a better value when wheat was sown after soybean (86.2%) than after winter oilseed rape (85.1%), as well as under CT conditions (87.1%) relative to the NT system (84.2%).

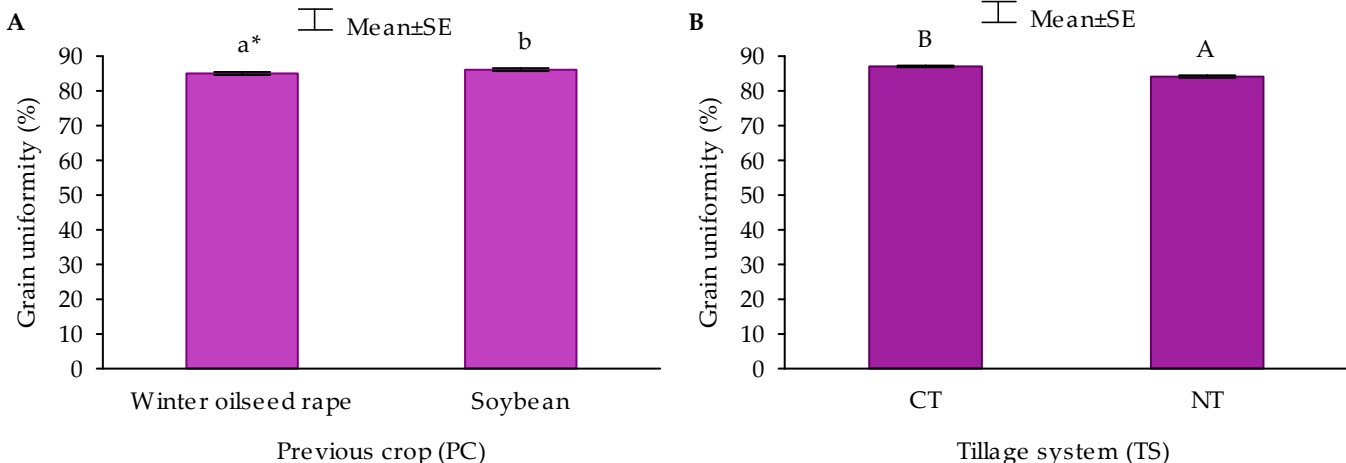

**Figure 10.** Uniformity of winter wheat grain depending on previous crop and tillage system (mean for 2014–2017). CT—Conventional tillage; NT—No-tillage. * Different letters denote significant differences ($p \leq 0.05$). The lowercase letters refer to previous crops (PC); the capital letters refer to tillage systems (TS).

The interaction of the previous crop and tillage system substantially determined the gluten index of winter wheat grain (Table 6). The highest value of this characteristic was shown for wheat grown in the field after soybean under NT conditions, and a similar value was found after the previous winter oilseed rape crop in the NT system. Compared to these treatments, a significantly lower value of the gluten index was determined after both previous crops under the CT system. The lowest value of the trait in question was found in the field after soybean under CT. The statistical analysis did not confirm the interaction of the experimental factors (PC and TS) in determining the other quality parameters of winter wheat grain.

**Table 6.** Interactive dependencies of previous crop and tillage system in determining the quality parameters of winter wheat grain (mean for 2014–2017).

| Specification | Wheat after Oilseed Rape | | Wheat after Soybean | | *p*-Value |
|---|---|---|---|---|---|
| | CT | NT | CT | NT | |
| Total protein content (%) | 12.6 a [1] | 12.6 a | 13.2 a | 12.8 a | 0.1183 |
| Wet gluten content (%) | 28.2 a | 28.0 a | 28.6 a | 28.4 a | 0.9795 |
| Gluten index (%) | 80.2 b | 82.8 c | 76.2 a | 84.2 c | 0.0000 *** |
| Zeleny sedimentation value (mL) | 33.6 a | 34.8 a | 37.1 a | 35.2 a | 0.1211 |
| Grain test weight (kg hL$^{-1}$) | 77.2 a | 76.4 a | 77.6 a | 76.8 a | 0.9739 |
| Grain uniformity (%) | 86.8 a | 83.4 a | 87.4 a | 85.0 a | 0.1235 |

CT—Conventional tillage; NT—No-tillage. [1] Different letters denote significant differences ($p \leq 0.05$). The same letter means not significantly different values (values for horizontal lines). *** significance level at $p \leq 0.001$.

## 4. Discussion

This study demonstrated that sowing winter wheat after soybean, compared to the stand after winter oilseed rape, significantly increased grain yield, as well as all the yield and crop components evaluated. The positive effect of previous legume crops on wheat yield was also proven by Plaza-Bonilla et al. [15] who obtained a higher grain yield after a legume crop (pea (*Pisum sativum* L.)) than after an oilseed crop (sunflower (*Helianthus annuus* L.)). In the opinion of these authors, legume crops reduce the requirement for

nitrogen fertilizers without detriment to subsequent yields of cereal crops and grain quality. On the other hand, a study by Sieling and Christen [11] found a higher grain yield after a previous pea crop than where winter wheat was sown after itself. In the authors' opinion, this could have been due to the higher incidence of diseases after the previous cereal crop and the slow decomposition of wheat straw residue, hence the lower availability of N for the succeeding crop. The beneficial effect of growing wheat after a previous soybean crop can undoubtedly be attributed to the fact that legume crops are able to fix nitrogen thanks to their symbiosis with nodule bacteria. A part of nitrogen enters the soil with legume crop residue and after mineralization becomes available for succeeding crops [13,42]. In the opinion of Angus et al. [14], the average increase in yield after legume crops, compared to growing wheat after wheat, is 1.2 Mg ha$^{-1}$. The beneficial influence of legume crops on wheat yield is associated, among others, with an increase in the number of ears per 1 m$^2$ [16]. Our research also showed that the positive influence of the previous soybean crop on winter wheat yield was predominantly due to the higher ear density.

The presented study revealed that the tillage systems (TS) did not cause significant differences in grain yield of winter wheat. A significantly higher number of ears per m$^2$, but lower TGW, was found in the CT system compared to the NT. This resulted in a small differentiation of the grain yield between CT and NT, because in both systems, the grain yield was significantly dependent on the number of ears per 1 m$^2$, and in NT also on the weight of 1000 grains. The studies of other authors demonstrate that the NT system, in comparison with CT, can have a significant effect on decreasing the grain yield of cereal crops [10,22,43]. Grigoras et al. [44] proved that the NT system causes a lower yield of wheat grain by about 14% compared to conventional tillage. Opposite results were obtained by Ali et al. [12] in southern Italy where in their experiment, the best production effects were achieved when durum wheat (*Triticum turgidum* L. *var. durum*) was grown under the NT system. In such a case, the grain yield was significantly higher than under CT (about 14%), whereas the lowest grain yield was obtained for reduced tillage (RT)—lower by 7.8% than in CT and by 19.0% than in NT. Thus, the present study and the studies of other authors' yield reveal that yields of cereal crops grown under different tillage systems largely depend on habitat and weather conditions, as well as on the crop species. In a study by De Vita et al. [21], a strong correlation was observed between yield and rainfall during the wheat growing season. In these authors' opinion, the beneficial effect of NT on crop yield can be noticed in years with a low amount of rainfall. In our experiment, a sufficient amount of rainfall was observed during the period of intensive grain development and filling in all growing seasons of winter wheat. Although, during the 2014/2015 growing season, the total precipitation was lower than the long-term average for this period, but the least precipitation occurred in February and in the grain harvest month (August), and therefore, it was not of major importance for wheat growth and yield. In the opinion of Castellini et al. [45], the positive effect of NT on the yield of crops, including winter wheat, can usually be observed only after long-term use of this tillage system. After the transition from conventional tillage to no-tillage, soil deterioration may initially occur. Chandrasekhara et al. [46] discussed model approaches to studying soil porosity during the transition from CT to NT. In these authors' opinion, after the transition from the CT system to NT, a deterioration in soil properties, including decreased soil porosity, may last up to about 4–5 years. However, there is a gradual improvement in soil properties; among others, the soil air and water regime improves, the soil structure stabilizes, and there is an increase in organic carbon content, as well as in the activity of microflora and microfauna [47–49]. Moreover, according to Woźniak and Gos [22], the NT system increases the content of total N and available phosphorus, as well as the number and weight of earthworms (*Lumbricus terrestris* L.) in the soil compared to CT. The improvement in soil physical, chemical, and biological properties under no-tillage conditions results in better soil water and air availability and enhanced growth conditions, consequently providing greater crop yields. In our study, the tillage systems did not cause differences in grain number and weight per ear. Conversely, Jung et al. [43] proved that growing winter

wheat in the NT system resulted in a decreased number of grains per ear. Woźniak and Rachoń [10], on the other hand, demonstrated based on a three-year field study, that lower yield of winter wheat under the NT system compared to that harvested under CT could be attributable to a lower number of ears per 1 m². Kraska et al. [50] showed a similar relationship for spring wheat.

The positive effect of the previous soybean crop and CT on the straw yield of winter wheat revealed in our research has also been confirmed by studies of other authors. Sieling and Christen [11] found a positive effect of previous legume (pea) crop on straw yield, while Ali et al. [12] proved the CT system to have a positive impact as, under NT, they obtained a lower weight of wheat straw by 9.6%. In our research, the lower straw yield in NT than in CT resulted mainly from a lower plants density per unit area and shorter height of plants. According to Soane et al. [48], Morris et al. [51], and Arvidsson et al. [52], the reduction in the straw yield in the NT system as compared to the CT system may be due to the higher density and compactness of the soil, which may result in fewer plant emergences, slower growth rate, and the production of less aboveground biomass.

The statistical analysis did not show an interaction between the previous crop and tillage system in determining winter wheat grain yield. Nonetheless, Ali et al. [12] show a significant effect of the interaction between the previous crop and tillage system on wheat yield. In these authors' study, the positive influence of tillage ($p \leq 0.05$) on grain yield, yield components, and quality parameters, particularly under NT, was more evident when a legume (faba bean (*Vicia faba* var. *equina* Pers.) was a previous crop.

The study results demonstrate that the selection of previous crops has a significant effect on determining the quality characteristics of winter wheat grain. After the previous soybean crop, the grain protein content, Zeleny sedimentation value, and grain uniformity were found to be higher. The amount of protein in wheat grain is largely determined by the soil content of nitrogen, which is supplied not only through fertilization, but also comes from mineralization of cover crop residue. Legume crops, due to their symbiosis with nodule bacteria and because they leave a large amount of nitrogen-rich crop residue, can substantially enhance the nitrogen availability to succeeding crops. In this way, they contribute to an increase in protein content in winter wheat grain [13,42]. A reduced amount of protein in wheat grain also significantly affects gluten quantity and quality, thus contributing to a deterioration in the technological value of grain and its suitability for bread making [53]. A study by Jaskulska et al. [5] proved that all quality characteristics of grain and flour associated with nitrogen and protein content have a higher value when wheat is better supplied with nitrogen. In such a case, grain contains much more protein and gluten, as well as has a greater sedimentation value, while flour is characterized by high water absorption capacity and protein content.

The results presented in this paper reveal that the Zeleny sedimentation value, as well as the protein and gluten content in winter wheat grain, was similar for the CT and NT systems. Kraska et al. [50] did not show tillage systems (CT and NT) to have an impact on the value of the above-mentioned quality parameters of wheat grain, either. Ali et al. [12] and Amato et al. [54], however, found a much lower protein content in wheat grain under NT compared to the conventional tillage (CT) system and reduced tillage (RT). Different results were obtained by Grigoras et al. [44]. In these authors' study, the average protein and gluten contents in grain obtained under the NT system were higher by, respectively, 0.15% and 0.67% relative to CT. The results of our experiment and the study by Woźniak and Rachoń [10] indicate a beneficial effect of CT on winter wheat grain uniformity. The presented results of this study and those of other authors' research do not provide a clear answer to which tillage system has a more beneficial effect on the grain quality characteristics, because wheat grain quality largely depends on habitat conditions under which a study is conducted. This indicates the need to continue research under various soil and climatic conditions, including research on the long-term effects of tillage on crop quality.

## 5. Conclusions

The present study has proven that soybean is a better previous crop for wheat than winter oilseed rape. This is evidenced by the significantly higher grain and straw yield, as well as by the better-quality parameters of winter wheat grain: Grain protein content, Zeleny sedimentation value, and grain uniformity. After the previous soybean crop, ear density per 1 m$^2$, plant height, grain number and weight per ear, and thousand grain weight were also more favorable. The previous oilseed rape crop only increased the gluten index value. Grain yield of winter wheat was significantly positively correlated with ear density to the greatest extent, after both the previous soybean and winter oilseed rape crops. Under the CT and NT systems, grain yield as well as grain number and weight per ear were found to be similar. Compared to NT, the CT system increased straw yield, number of ears per 1 m$^2$, and plant height of winter wheat. Thousand grain weight, on the other hand, was greater for the NT system. Relative to NT, under CT conditions, grain test weight and uniformity reached much higher values, whereas gluten index had a significantly lower value. In the case of the CT system, wheat grain yield was most positively correlated with ear density, whereas under NT, it was with grain weight per ear. Among the evaluated yield and crop components, the interaction of the previous crop and tillage system significantly affected only the number of grains per ear and thousand grain weight. The lowest value for both these traits was found after the previous winter oilseed rape crop under NT and CT conditions. After both previous crops of winter wheat, NT significantly increased gluten index compared to CT.

**Author Contributions:** The authors contributed to this article in the following ways—conceptualization: D.G.; data curation: D.G. and M.H.; formal analysis: D.G. and M.H.; funding acquisition: D.G.; investigation: D.G. and M.H.; methodology: D.G.; project administration: D.G.; supervision: D.G. and M.H.; writing—original draft: D.G. and M.H.; writing—review and editing: D.G. and M.H. All authors have read and agreed to the published version of the manuscript.

**Funding:** Research supported by the Ministry of Science and Higher Education of Poland as the part of statutory activities of Department of Herbology and Plant Cultivation Techniques, University of Life Sciences in Lublin.

**Institutional Review Board Statement:** Not applicable.

**Informed Consent Statement:** Not applicable.

**Data Availability Statement:** The data presented in this study are available on request from the corresponding author.

**Conflicts of Interest:** The authors declare no conflict of interest.

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
