# Peer review of "Grain Yield and Quality of Winter Wheat Depending on Previous Crop and Tillage System"

_agriculture, doi:10.3390/agriculture11020133_

Round 1
Reviewer 1 Report
Authors study the effects of previous crops (soybean and winter rapeseed) and tillage systems (conventional tillage and no-tillage) on yield and quality parameters of winter wheat grain. Measurement data were collected in a four year long, four-field crop rotation of soybean – winter wheat – rapeseed – winter wheat.
Introduction is plenty with relevant publications and very well-built.
In the material and methods section soil, weather and management descriptions are appropriate.
However, in the experimental setup, it would be preferable to know how many years of no-tillage had been completed already at the experimental site. The authors refer to the study of Castellini et al. [44] in line 372. They point out that the positive effect of no-tillage on yield of crops can usually be observed only after long-term use of this tillage system (4-5 years). Could you please, evaluate your findings according to this statement?
Another point regarding material and methods that Latin names of the wheat, soybean and rapeseed species are missing.
Results are clearly presented. Tables and Figures are well organized and can be clearly followed.
My other main concern is that the discussion of the results could have been more detailed. I was curious about the possible reasons behind no-tillage caused straw yield decrease, while grain yield is almost the same as grain yield of conventional tillage (Figure 1. and 3.).
The conclusions of the work revealed the positive impacts of soybean on subsequent winter wheat growth and quality components. The study also highlighted that the gluten index of winter wheat grain was higher in no-tillage cultivation. However, grain weight and uniformity increased in conventional tillage system.
Specific comments:
Line 158. Please, consider using g m-2 for presenting results from a 32 m2 plot instead of per hectare basis.
Line 93. ’In the first growing season of winter wheat (2013/2014)’, however, the first season of the experiment was soybean in line 128. Please, clarify this issue.
Line 47. Is It sure that ‘grown on farmyard manure’ is right?
Line 59. 364. Consider rephrasing of ’crop plant species’.
Line 131., 288. Consider rephrasing the sentence.
Line 135. Consider adding the BBCH scale reference in the text of MM.
Line 357., 361., 397. Consider presenting the difference in percent between yields of NT and CT treatments. That would enhance understanding.
Author Response
Replies to the comments of Reviewer I
We greatly thank for the in-depth reviews and comments, thanks to which our paper has become more transparent and enriched with a new content that has contributed to its enhanced technical quality.
Replies to the comments of Reviewer I
- In MM chapter information, how many years no-tillage has been made at the experimental site was added.
- In MM chapter Latin names of the wheat, soybean and rapeseed species were added.
- In Discussion chapter the effect no-tillage on straw yield was explained.
Replies to specific comments of Reviewer I:
- Line 158. In our opinion, there is no need to change the units to g m-2, because those used in the manuscript are given in most scientific publications.
- Line 98. In the experiment all plants of crop rotation were conducted in all the fields simultaneously. In 2013 winter wheat and rape rapeseed were sown and in spring 2014 soybean was sown.
- Line 47. The sentence was corrected to “The potatoes and legumes are also appropriate previous crops for wheat [11, 12]”.
- Lines 59 and 364. The phrase ’crop plant species’ was corrected to ’crop species’
- Lines 131 and 288. In our opinion, the sentence in line 131 is correct. The sentence in line 288 was corrected to “In all research objects gluten can be assessed as strong”.
- Line 135. BBCH scale was added in the text of MM.
- Line 357, 361, 397 Following the Reviewer’s suggestion difference in percent between yields of NT and CT treatments were presented.

Reviewer 2 Report
Agriculture
REVIEWER COMMENTS
MS No. Agriculture-1086789
Title: Grain yield and quality of winter wheat depending on previ-2 ous crop and tillage system
COMMENTS
The manuscript is argued on a study concerning the crop rotation and soil tillage. The objective was to evaluate the effects of two previous crops (soybean and winter rapeseed) and of two soil tillage (conventional tillage and no-tillage) on yield and some parameters of winter wheat grain.
The study topic is surely very much interesting from both agronomic and scientific point of view.
The manuscript is well structured and organized, according to the journal directions (Introduction; Material and methods; Results, Discussion, Conclusions).
In the reading of the text, I found it pretty well written and arranged in the description.
Nevertheless, I found a flaw in the statistical analysis for which I have some remarks, reported below.
In Material and Methods.
Concerning the statistical analysis, the authors only say " The study results collected over the period 2014–2017 were analyzed by analysis of 171 variance (ANOVA), while the significance of differences was estimated by Tukey’s test...".
I have to highlight that they don't specify the adopted model.
Because there is the year involved in the analysis, it is necessary to describe the adopted statistical model, which must consider the year as repeated measure. In these cases I use to suggest the use of mixed model or a split-plot appropriate model in which the year is a repeated measure. Alternatively, if they know can use an adequate model to run in which the year is considered repeated measure.
Results
The data reported in the tables 1 and 2 show different trend of climatic factors during the experimental period. In all reported data (figures and tables), the authors never highlighted the difference due to the time (years). This is a little bit strange fact. In my opinion adopting a correct model the year will result significant in the statistical analysis, consequently it will be necessary report the data differently. In fact, the authors report also in the text the evidence of the year effect even if in the tables and figures it is not distinguished.
In the tables and figures caption it is not specified if the letters for distinguishing the significantly different values involve all treatments or not. The comparisons using the letters should be within the same treatment. So a good solution which can be adopted in the figures and tables reported in the manuscript is the use of capital letters for a treatment and lowercase letters for the other treatment, within the same figure.
So, I suggest that these improvements of the manuscript are needed.
Author Response
Replies to the comments of Reviewer II
We greatly thank for the in-depth reviews and comments, thanks to which our paper has become more transparent and enriched with a new content that has contributed to its enhanced technical quality.
Replies to the comments of Reviewer II
- In MM chapter was described the adopted statistical model, which was used in the analysis of variance and was explained why research years were not shown (lines….).
- In the figures statistical differences were marked by the lowercase letters for previous crops (PC) nad the capital letters for tillage systems (TS).
Best regards
